# Commercial hatchery practices have long-lasting effects on laying hens' spatial behaviour and health

Camille M. Montalcini [1,2]*, Matthew B. Petelle[1], Michael J. Toscano[1]

**1** ZTHZ, Division of Animal Welfare, VPH Institute, University of Bern, Zollikofen, Switzerland, **2** Graduate School of Cellular and Biomedical Sciences, University of Bern, Bern, Switzerland

\* camille.montalcini@unibe.ch

**Data Availability Statement:** The data and code for this study are available on https://doi.org/10.17605/OSF.IO/5WQGD.

**Funding:** M.J.T. received funding from the Swiss National Science Foundation (grant number

## Abstract

The commercial hatchery process is globally standardized and exposes billions of day-old layer chicks to stress every year. By alleviating this early stress, on-farm hatching is thought to improve animal welfare, yet little is known about its effects throughout production. This study compared welfare indicators and spatial behaviours during the laying period of hens hatched in an on-farm environment (OFH) to those hatched in a commercial hatchery and transferred at one day-old to a rearing barn (STAN). In particular, we assessed how OFH and TRAN hens differed in space-use and movement behaviours following the transfer to the laying barn at 17 weeks of age, a similar stressor encountered by STAN hens early in life, and determined whether effects aligned more with the 'silver-spoon' or 'environmental matching' hypothesis. We found that for the first three months post-transfer into the laying barn, OFH hens, on average, transitioned less between the aviary's tiers and spent less time on the littered floor. Because OFH hens became behaviourally more similar to STAN hens over time, these results suggest that OFH hens required a prolonged period to establish their daily behavioural patterns. Furthermore, OFH hens had more severe keel bone fractures throughout the laying period but similar feather damage and body mass to STAN hens. No differences were found in hen mortality or the number of eggs per live hen. These findings support the environmental matching hypothesis and suggest that early-life stressors may have prepared hens for later-life stressors, underscoring the importance of both early-life and adult environments in enhancing animal welfare throughout production.

## Introduction

Every year, billions [1] of day-old layer chicks are exposed to stress [2] because of the globally standardized commercial hatchery process [3]. Chicks in hatcheries, are subjected to loud noise during incubation (~90dB), hatched in darkness, prevented from accessing feed, water, or litter, and subjected to sexing, vaccination, and transportation to rearing farms at one day of age [2]. The early exposure to stressors can be alleviated by hatching chicks on farm, where chicks are transported before hatching and have direct access to feed, water and litter.

310030_189056) and the European Union's Horizon 2020 research and innovation programme (grant agreement N°101000236). The document reflects only the author's view and the European Union's Horizon 2020 research, and innovation programme is not responsible for any use that may be made of the information it contains. This project is part of EuroFAANG (https://eurofaang.eu). The funders had no role in study design, data collection and analysis, decision to publish, or preparation of the manuscript.

**Competing interests:** The authors have declared that no competing interests exist.

Exposure to less stressful environmental conditions during development may have a positive impact on various aspects of individual fitness later on, a phenomenon called "silver-spoon" effect [4] (for a review on birds and mammals, see [5]). The silver-spoon hypothesis suggests that individuals who experience relatively better environmental conditions during development, such as characterized by reduced stressors or abundant nutritional availability, may be able to cope better when confronted with adversity later in life. Accordingly, it is not surprising that on-farm hatching has been shown to improve the well-being of broilers and laying hens later in life, in particular throughout development. Specifically, this practice has been shown to reduce total mortality [5] and footpad dermatitis [6, 7] of broilers, and increase body mass [3, 8, 9] and reduce feather damage, comb injuries, and corticosterone reactivity during restraint [2] of laying hens.

Alternatively, by experiencing adverse environments during development, one may be prepared or adapted to handle similar adversities in the future. The "environmental matching" hypothesis suggests that environmental conditions in early life shape an individual phenotype via developmental plasticity [10], so that an individual is adapted to similar environmental conditions experienced earlier in life [11, 12]. Therefore, chicks that hatched in a commercial hatchery and subsequently transported on farm could have a phenotype more adapted to aversive environments, such as transportation to a new environment, than chicks hatched on-farm. Thus, the benefit of a less aversive early life environment would depend on the environmental conditions experienced later in life [11–13]. However, studies evaluating the effect of the commercial hatchery process on adult laying hens in commercial settings are scarce, limiting our understanding of their long-term effects on hen welfare. The limited understanding is especially true for health issues that predominantly arise during adulthood or may worsen as hens age, such as feather damage [14, 15] or keel bone fractures (KBF) [16, 17]. Thus, a long-term approach is necessary to understand whether on-farm hatching improves hen welfare throughout production, or whether its relative benefits are eventually offset by later stressors that they are unable to manage.

In this study, we compared the severity of KBF, feather damage, body mass and spatial behaviours of laying hens hatched on-farm (OFH) to those hatched in a commercial hatchery and transferred at one day of age to the farm (STAN). Our goal was to determine if, and for how long, these two different environments experienced at one day of age could account for variations in animal welfare and behaviour during the laying period. We assessed how the transfer from the rearing to the laying barn, a similar stressor encountered by day-old STAN chicks, affected space-use and movement behaviours, as well as welfare indicators. We aimed to determine whether the observed effects aligned more with the 'silver-spoon' or 'environmental matching' hypothesis. The former hypothesis would be supported if OFH hens would display overall greater welfare compared to STAN hens, while the latter hypothesis would be supported if OFH would display overall worse welfare conditions. Better welfare could be here manifested via less severe KBF, reduced feather damage, and greater time spent in the littered floor and the winter garden. These areas provide enhanced opportunities for the expression of natural behaviours such as locomotion, exploring, foraging, scratching, and dust bathing, which are important for laying hen welfare [18, 19].

## Materials and methods

### Ethical note

The research was conducted in accordance with the cantonal and federal regulations for the ethical treatment of experimentally used animals. All procedures were approved by the Bern Cantonal Veterinary Office (BE-45/20).

## Study design

All Dekalb white chicks (*Gallus gallus domesticus*) originated from the same parent flock and began incubation off-site using standard hatchery practices. At 18 days of development, three days before hatching, 3,300 eggs were arbitrarily chosen as part of the on-farm hatch (OFH) treatment and all except 270 clear eggs transported to a commercial rearing barn at the Avi-forum facilities in Switzerland. The eggs were transported in a commercial vehicle for less than 1.5 hours, which maintained a stable environmental temperature at an average of 36.4°C. Eggs were positioned in HatchTech Setter Trays 15 cm above the littered floor, where feed, water, and litter were available. We monitored environmental conditions and temperature of 30 eggs every six hours throughout the hatching process. Specifically, we ensured that the ambient relative humidity remained above 30%, the windspeed below 0.15 m/s, ambient temperature above 32°C, and the eggshells temperature between 35–38°C (see S1 Fig for eggshell temperature over time). Our methodology is similar to a previous study conducted on layer chicks [9].

At one day of age (DOA), OFH chicks were manually sexed by examining their wing feathers for sex-specific patterns and females were vaccinated (IB 4/91). On the same day, 1,200 chicks from the commercial hatchery were transported to the rearing barn as part of the STAN treatment. Transportation took place in a commercial vehicle over a duration of eight hours, during which a consistent environmental temperature around 28°C was maintained. Unlike the OFH chicks, STAN chicks—in addition to being transported to new housing—hatched in darkness and were deprived direct access to food, water, or litter after hatching until arrival at the rearing barn. Similar to OFH chicks, STAN chicks were vaccinated and manually sexed by examining their wing feathers for sex-specific patterns at the hatchery facility. Although both OFH and TRAN chicks were manually sexed by the same company managing the post-hatch procedures (Prodavi SA, CH), we supervised the sorting of OFH chicks and encouraged the sexers to proceed gently, however no objective comparison was made of the handling procedures between treatment groups. STAN chicks were used to populate two rearing pens and OFH chicks were used to populate the other two pens (600 hens/pen). Males and surplus females were returned to the hatchery for humane disposal. At seven DOA all chicks were classified into a more/less explorer class. We did not use the class as an exploratory behaviour as the measurement could not be validated (S1 Text), though we controlled for the class in subsequent analysis. Simultaneously, 160 focal birds (40 hens/rearing pen) were selected from the 2'400 chicks. Of the 160 focal birds, 80 focal birds were classified as MEXP or LEXP (40 / class), while 80 were selected as a representative sample and used for another study that collected brain tissues throughout the laying period.

At 119 DOA, all hens were caught, put into a crate, and transported to one of eight laying pens on the same site (225 hens/pens, including 20 focal hens, four pens/treatment). Bird density was 8.1 hens per square-meter of permanent accessible area (225 hens/27.92 $m^2$). The laying barn contained a quasi-commercial multi-tier aviary system (Bolegg Terrace separated into 20 pens by grids illustrated in S2 Fig; indoor length x width x height until the top tier grid floor: 7 x 2.3 x 2.69 m; previously described [20]) and an outside covered winter garden (WG; 9.32 $m^2$) accessible by pop holes from 10:00 h to 16:00 h on most days. On the day of transfer to the laying barn the 160 focal hens were assigned a tracking device to continuously register their transitions across the indoor aviary levels and the winter garden (WG) until near the end of production (tracking period: September 2020 –July 2021). At five time points during the laying period (DOA: 127, 173, 243, 313, and 418), 16 randomly selected focal hens were killed (eight hens/treatment) to collect brain tissues as part of a separate study. Each of these time points also included welfare assessment (described in the below section), except for DOA 127, which was replaced by DOA 215 to capture more variation in animal welfare. For each hen

killed, another hen from the same pen was arbitrarily selected to continuously track the same number of hens, for a total of 227 hens used in the study.

## Welfare indicators

Welfare assessment included feather damage, radiographs for KBF, and body mass (digital scale in grams). During the welfare assessment, the observers were blinded to the treatment, laying pen identity, and hen class, and shown reliable in a previous effort for both feather damage and KBF severity scores [21]. The feather damage score (continuous, 0–100) was assigned using the photographs of white laying hens which we rescaled to 0–100 and took the complement to 100 so that higher scores are indicative of poorer welfare (score 1: approx. 100–76 depending on the extent of damage; score 2: approx. 75–51; etc.) for each body part [22]. More precisely, we assigned a score of the breast, tail, and neck, but not the back and wing feathers as these could not be reliably assessed because of the backpack containing the tracking tag (described below). We then averaged these to get an overall individual feather damage score. We assessed KBF severity (continuous, 0–100) based on the latero-lateral radiographs using the scoring methodology described by Rufener et al. [23], where the score is described as an indicator of the total amount of keel bone affected by fractures. We excluded the first time-point of KBF severity and feather damage, as there was little variation, with both having a median score of zero.

## Spatial behaviours

We tracked individuals' transitions across five zones: the four different levels of the aviary (top tier, nestbox tier, lower tier, and littered floor) and the outside covered WG. We used a low-frequency tracking system with active tags (mass: 28.1 g) enclosed in a backpack mounted on the back of the hens (see Montalcini et al. (2022) [24] for the validation and description). Tracking data were collected from the first full day in the laying barn (DOA 119) until near the end of production (DOA 416). We excluded days with known disturbances (e.g., vaccinations or welfare assessments) and those with known tracking system malfunctions (e.g., low battery level). Subsequent analysis involved a period of 297 days of tracking, during which hens had on average 169 days tracked, with a minimum of three days tracked and a maximum of 250 days, involving a total of 227 hens and 38,303 hen-days observations.

   We characterized the daily movement and space-use behaviours of each hen with six behaviours expressed while artificial light was provided. We used the (i) vertical travelled distance, defined as the total number of indoor tiers crossed, to account for the level of vertical movement. We used the proportion of the indoor time spent on the (ii) top tier, (iii) nestbox tier, and (iv) littered floor to account for indoor space-use behaviours, and (v) WG presence (yes/no) to account for the outdoor space-use behaviour. Finally, because the nestbox tier is of particular interest within commercial settings, we also used the (vi) time when a hen reached half of its nestbox tier duration, accounting only for hours where hens are expected to lay, that is between 02:00h and 08:00h, hereafter referred to as the nestbox tier timing.

## Production traits

The female hatchability, i.e., the percentage of healthy female hatched, was 40% in the hatchery and 42.6% on-farm within a hatching window duration of 65 hours (for the on-farm chicks, see S1 Table for the hatching rate over time), was comparable to previously reported OFH results [9]. During the rearing period, there was a total of 11 deaths for each treatment (i.e., < 1%). We analyse production traits after the transfer to the laying barn, as hens had not laid eggs prior to that point. First, we used the number of early deaths per day in each pen during

the laying phase. This dataset is right censored where the value 1 represents death, and 0 indicates being alive. Second, we used the hen daily average (average number of nest eggs per live hen) in each pen. Throughout the laying period, eggs laid inside the nestboxes were collected consistently at the same time every day and counted at the pen-level. We did not include floor eggs in our analysis as they represent approximately 0.24% of total eggs laid. In addition to the four pens per treatment group (STAN and OFH) with focal birds, we also used the data of two additional pens without focal hens but containing 205 birds from one treatment group with an additional 20 Lohmann LSL hens that hatched in the hatchery (called "special pens").

## Statistics

**Welfare indicators.**　Statistical analyses were conducted in R version 3.6.1. To evaluate treatment effects on hen's welfare indicators, we fitted one linear mixed-effect model from the 'lme4' package per welfare indicator (body mass, KBF severity, and feather damage) as a function of date (or health assessment identity), treatment, and date-treatment interaction. We controlled for class and included hen identity nested in pen identity as a random term. Pen identity was removed when fitting the KBF severity and body mass due to low variance leading to convergence issues. We scaled body mass to be within 0–1 within each welfare assessment separately prior to the model fit. Model assumptions were checked visually (normality and homoscedasticity of residuals). To assess significance of the date-treatment interaction, we compared each model with a model that did not contain the interaction variable using the function Anova from R. When the date-treatment interaction was significant ($p < 0.05$), we reported results from a post-hoc analysis with adjusted p-values (Bonferroni adjustment, package "emmeans"). When the date-treatment interaction was not significant, we removed it from the model and assessed significance of the treatment as a main effect by comparing the full and reduced models.

**Spatial behaviours.**　To evaluate whether treatment groups differed in mean behaviours after the transfer to the laying barn, we fitted one generalized linear mixed-effect model from the package 'glmmTMB' [25] for each behaviour for the first month in the laying barn. To complement the findings and evaluate how long treatment groups differed in mean behaviours, we fitted models for each following month as well (10 months, six behaviours, total of 60 models), as a function of the treatment (with STAN as reference group). We chose month as the unit to analyse treatment effect over time, aiming to strike a balance between thoroughly estimating mean effects over time and avoiding potential noise linked to shorter time intervals like weeks. A previous study on the same hens, studying intra-individual variation in a composite behaviour, found that, on average, hens increased their indoor movements for 39 days after the transfer to the laying barn [21]. This previous result suggests that the transfer to the laying barn could have a long-term effect on hens' spatial behaviours. However, we expected that any treatment differences in spatial behaviours would appear directly after the transfer to the laying barn and diminish over time. Therefore, any statistically significant treatment effect that does not follow that pattern was interpreted with care, and the results section emphasized coefficient estimates rather than p-values. We controlled for the class, time (defined as the number of days since the transfer to the laying barn), KBF severity, body mass, and number of hours with artificial lights on, by including them as fixed effects. We interpolated linearly (with monotonically increasing) both the KBF severity score and body mass for each hen separately to better control for their health between two consecutive health assessments considering that both scores exhibit an upward trend over time. The hen identity nested in the pen identity was included as a random term. To avoid convergence issues due to the very low explained variance by the pen identity, we performed a likelihood ratio test with and without

the pen identity and chose the full model when the p-value was < 0.05. For the WG-related behaviour, we also controlled for the number of hours the WG was accessible and the average daily external temperature (˚C), taken from the LSZB weather station (~12 km from the barn) and accessed via the Wolfram alpha API in Python. All continuous variables were scaled by two times the deviation to obtain coefficients comparable to those of binary predictors (i.e., the treatment) [26]. As the first full day in the laying barn coincided with the final day of September, we incorporated that day into the models for the first month (October).

The vertical travelled distance was modelled with a gaussian family for months 2–10 and with a zero-inflated Poisson model with the rescaled number of days in the barn as the zero-inflation parameter for month 1, as model assumption were otherwise not met. The nestbox tier usage was modelled with a gamma family (with a log-link function) and the WG presence with a binomial family (with a logit-link function). The proportion of indoor time spent on the top and nestbox tiers were both modelled with a *beta* family (with a logit-link function). The proportion of the indoor time spent on the littered floor was modelled with a gaussian family for months 2–10 and with a binomial family (with a logit-link function) to account for the excess of zeros (19.7% of observations) for month 1. Behaviours used in a beta distribution were first rescaled between 0.01 and 0.99. During the first month in the laying barn, hens had not reached the peak of production and the artificial light was turned on later in the day (days 1–9 at 9h, days 10–16 at 8h, days 17–22 at 7h, days 23–24 at 6h, days 25–30 at 5h, days 31–32 at 4h). Therefore, we did not analyse the first month in the laying barn for the two nestbox tier related behaviours because behaviours expressed during that first month would likely not be comparable with behaviours expressed in subsequent months. In addition, we did not use the first seven days when fitting the WG presence, as the opening of the WG was delayed until normal laying behavior commenced (i.e., the eighth day) as is common for commercial practices. Residuals were simulated using the 'DHARMa' package to verify model assumptions (normality and homoscedasticity of the residuals). We reported the bootstrapped coefficients [27], credible intervals, and p-values computed from the 'parameters' package using 500 iterations.

**Production traits.**   To analyse daily mortality, we first estimated survival curves per treatments (without the special pens) and per pen (with the two special pens) using the Kaplan-Meier method with the 'survival' package in R [28]. Then, we used the log-rank test to test whether the treatment group (with the two special pens) differ in their survival curves. In order to account for the pen identities as a random effect and the "special pens" as a fixed effect, we also fitted a Cox proportional hazards models with both fixed and random effects, using the 'coxme' package in R [29]. Because it is a fundamental assumption of both the log-rank test and the Cox proportion hazard test that the hazard ratio is constant over time [30], we tested the proportional hazards assumption. We found that the effect of both the pen identity, special pen effect, and the treatment covariates were not time-dependant (p<0.05). However, because there was a trend (p = 0.06) suggesting that the effect treatment had on the time until death may not be constant over time, we also split the data into two sets [31], considering the first 60 days in the laying barn (when there is a peak of mortality) and the remaining production period. The effect of treatment no longer was time dependent, and the effect of treatment remained the same. Therefore, we reported the result from the model with the full rather than split time period.

We analysed the daily average number of eggs per live hen in each pen using a sigmoid curve, $\frac{a}{1+exp\left(\frac{m-time}{s}\right)}$, where *time* represented by the number of days since being in the laying barn, *a* as the horizontal asymptote, *m* as the time point at *a*/2, and *s* as the steepness of the curve at *a*/2 [32]. Sigmoid curves were fitted using a nonlinear least square algorithm (nls

function in R). We fitted a sigmoid curve for each pen separately and provided pen-level parameter estimates. More precisely, for each pen, we reported *a* as an indication of the level at which egg-production stabilises and *m* as an indication of the time point with the steepest slope, that is at the inflection point of the curve. We did so considering first the entire period and then only the onset of lay (first 60 days in the laying barn) as it is an important period in which we might observe the greatest variation between hens. Then, to statistically assess treatment effect we fitted a nonlinear mixed-effects model with the 'nlme' package [33] with treatment as fixed effect and pen identity as random effect on all three parameters (a, m, s). We did so when considering the entire period in the laying barn as well as when considering solely the first 60 days.

## Results

### Welfare indicators

The date-treatment interaction was neither a predictor of body mass ($x^2_{6,1121} = 9.49, p = 0.15$), feather damage ($x^2_{2,506} = 4.95, p = 0.08$), or KBF severity ($x^2_{3,669} = 4.34, p = 0.23$) and was therefore removed from all models. Treatment as a main effect was neither a predictor of body mass ($x^2_{1,1121} = 1.95, p = 0.16$) or feather damage ($x^2_{1,506} = 0.54, p = 0.46$), but was a predictor of KBF severity ($x^2_{1,669} = 6.35, p = 0.01$). Specifically, holding all else equal, the KBF severity score of OFH hens was, on average, 4.53 ([95% CI] = [0.99, 8.07]) points higher than that of STAN hens. Estimated marginal means (±95% CI) are presented in the S2 Table and model estimates and p-values are presented in S3 Table. The observed scores of each welfare indicator per treatment and date are illustrated in a violin plot (Fig 1), and their means (±SD) presented in Table 1.

### Spatial behaviours

Model coefficients from the treatment predictor for each behaviour across months are displayed in Fig 2 (bootstrapped p-values and coefficients are detailed in S4 Table). To help further interpret the nature of the change, particularly in cases where a treatment effect was found for several months only (e.g., determining which, if any, of the treatment group had their behavioural responses *converging* toward those of the other group), the mean observed behavioural scores are presented per treatment and month in Table 2.

We can interpret exponentiated coefficients from our models with a beta family (i.e., models fitting the proportion of time spent on the top tier and the nestbox tier) as odds ratios. For

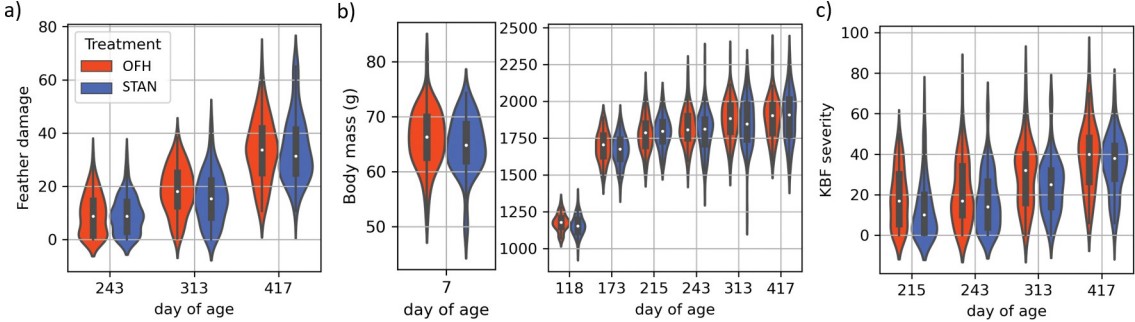

**Fig 1. Violin plot of the raw welfare indicators over day of age per treatment.** Red and blue colours represent the treatment groups, OFH and STAN, respectively, and feather damage is represented in (a), body mass in (b), and KBF severity in (c).

**Table 1.  Mean (±SD) observed values of raw welfare indicators per treatment groups.**

|  | Feather damage | | Body mass (g) | | KBF severity | |
| --- | --- | --- | --- | --- | --- | --- |
| Day of age | OFH | STAN | OFH | STAN | OFH | STAN |
| 7 |  |  | 66.04 ±5.17 | 64.75 ±5.12 |  |  |
| 118 |  |  | 1177.75 ±57.88 | 1157.16 ±64.90 |  |  |
| 173 |  |  | 1692.88 ±103.13 | 1671.03 ±99.26 |  |  |
| 215 |  |  | 1780.19 ±118.01 | 1790.72 ±105.09 | 18.35 ±14.16 | 12.72 ±14.57 |
| 243 | 9.58 ±7.73 | 9.24 ±7.02 | 1822.38 ±122.70 | 1795.32 ±137.46 | 21.50 ±16.21 | 16.86 ±13.98 |
| 313 | 18.57 ±8.92 | 15.83 ±9.59 | 1877.29 ±134.15 | 1855.84 ±160.46 | 30.20 ±16.26 | 24.57 ±13.68 |
| 417 | 33.60 ±11.79 | 34.09 ±12.31 | 1888.44 ±139.77 | 1898.60 ±168.97 | 38.64 ±16.34 | 34.94 ±14.25 |

example, during the first month in the laying barn, the odds of being on the top tier for OFH hens was 1.49 times that of STAN hens (95% CI [1.12, 2.04], p = 0.01; Fig 2A), indicating that on average OFH hens spent more time on the top tier. A similar effect was found for month 2 (β [95% CI] = 1.67 [1.14, 2.35], p < 0.01). Similar interpretations can be made for models with a proportion of time spent on the top tier or nestbox tier as response variables, although neither models related to the nestbox tier (Fig 2C and 2F) nor to the top tier (Fig 2A) had a significant effect of treatment beyond the second month.

We can interpret the exponentiated coefficients from our models with a binary response (i.e., models fitting the proportion of time spent on the littered floor (during month 1) and the WG presence responses) as odds ratios. For instance, during the first month in the laying barn, our results indicate that OFH hens were less likely to go to the littered floor compared to STAN hens. More specifically, we found an odds ratio of 0.30 (95% CI [0.12, 0.67], p < 0.01),

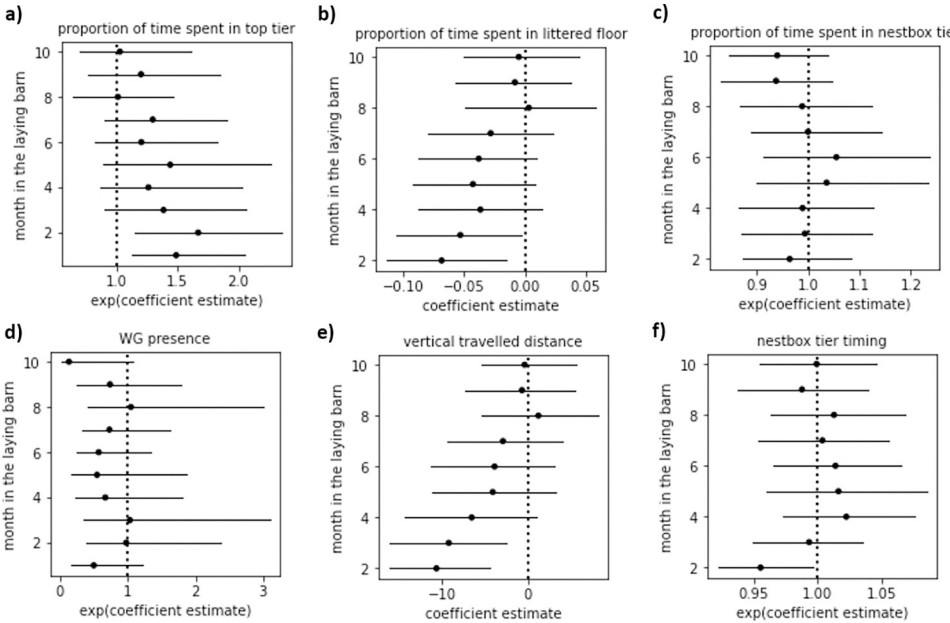

**Fig 2. Coefficient estimates of treatment (with STAN as reference group) for each behaviour across months after the transfer to the laying barn.** We interpreted significance whenever the confidence interval did not cross the dashed line. If the point and confidence interval lie to the right of the dashed line, it indicates that the model estimates higher behavioural response in OFH hens compared to STAN hens. Bootstrapped outputs (estimates and p-values) are detailed in S4 Table.

**Table 2. Mean observed values of raw behavioural data per treatment groups.** We highlighted in bold the months and behaviours for which we found a significant treatment effect.

| month in laying barn | proportion of indoor time spent | | | WG presence | vert. travelled distance | nestbox tier timing |
|---|---|---|---|---|---|---|
| | top tier | littered floor | nestbox tier | | | |
| month1 | **0.53 *vs.* 0.42** | **0.26 *vs.* 0.32** | | 0.44 *vs.* 0.51 | **28.51 *vs.* 39.02** | |
| month2 | **0.41 *vs.* 0.30** | **0.31 *vs.* 0.39** | 0.07 *vs.* 0.07 | 0.62 *vs.* 0.65 | **47.03 *vs.* 58.67** | **5.00 *vs.* 5.31** |
| month3 | 0.34 *vs.* 0.28 | **0.36 *vs.* 0.41** | 0.07 *vs.* 0.07 | 0.66 *vs.* 0.67 | **55.43 *vs.* 63.99** | 4.24 *vs.* 4.29 |
| month4 | 0.33 *vs.* 0.27 | 0.38 *vs.* 0.42 | 0.08 *vs.* 0.08 | 0.62 *vs.* 0.69 | 61.64 *vs.* 68.68 | 4.19 *vs.* 4.14 |
| month5 | 0.33 *vs.* 0.28 | 0.37 *vs.* 0.42 | 0.08 *vs.* 0.08 | 0.61 *vs.* 0.70 | 63.03 *vs.* 68.94 | 4.16 *vs.* 4.18 |
| month6 | 0.30 *vs.* 0.24 | 0.40 *vs.* 0.44 | 0.09 *vs.* 0.09 | 0.65 *vs.* 0.73 | 66.50 *vs.* 72.47 | 4.46 *vs.* 4.52 |
| month7 | 0.31 *vs.* 0.28 | 0.40 *vs.* 0.41 | 0.09 *vs.* 0.09 | 0.62 *vs.* 0.69 | 61.38 *vs.* 65.76 | 4.59 *vs.* 4.62 |
| month8 | 0.30 *vs.* 0.29 | 0.41 *vs.* 0.41 | 0.08 *vs.* 0.08 | 0.69 *vs.* 0.72 | 58.36 *vs.* 58.61 | 4.78 *vs.* 4.74 |
| month9 | 0.28 *vs.* 0.26 | 0.43 *vs.* 0.43 | 0.08 *vs.* 0.09 | 0.71 *vs.* 0.83 | 57.11 *vs.* 58.38 | 5.05 *vs.* 5.18 |
| month10 | 0.27 *vs.* 0.26 | 0.43 *vs.* 0.43 | 0.08 *vs.* 0.08 | 0.68 *vs.* 0.80 | 52.95 *vs.* 53.14 | 5.34 *vs.* 5.36 |

meaning that the odds of going on the littered floor at least once during the day (during the first month) in OFH hens were 0.30 times the odds in STAN hens. A similar effect was maintained during months 2 and 3 (with a gaussian family; Fig 2B). More specifically, compared to STAN hens, OFH hens spent 7% and 5% less of their daily indoor time on the littered floor during months 2 and 3, respectively (month 2: β [95% CI] = -0.07 [-0.11, -0.02], p = 0.008; month 3: β [95% CI] = -0.05 [-0.11, -0.00], p = 0.04). Treatment was not a significant predictor of WG presence (see Fig 2D).

Furthermore, we found that OFH hens moved less vertically during the first three months (see Fig 2E). More specifically, hatching on farm was associated with a reduction of 31% in the vertical travelled distance during month 1 (exp(β) [95% CI] = 0.69 [0.59, 0.82], p < 0.001, Poisson distribution). The treatment effect persisted up to month 3, with OFH hens crossing on average 11 and 9 fewer zones per day than STAN hens during months 2 and 3, respectively (month 2: β [95% CI] = -10.62 [-16.13, -4.35], p < 0.001; month 3: β [95% CI] = -9.19 [-16.13, -2.50], p = 0.012, Gaussian distribution).

Lastly, OFH hens, on average, were slightly earlier in their nestbox tier timing during the second month than STAN hens (β [95% CI] = 0.96 [0.92, 0.99], p < 0.05; Fig 2F). The treatment showed no effect in subsequent months in the nestbox tier timing, and we found no effect of treatment on the proportion of time spent on the nestbox-tier.

## Production traits

The survival probability for each treatment with 95% CI and per pen are represented in Fig 3A–3C, respectively. Results from the log-rank test revealed no statistically significant difference in the survival curves between the two groups is ($x^2$ = 1.5, df = 1, p = 0.20). Similarly, results from the Cox proportion hazard model revealed no statistically significant difference in the hazard between the treatment groups (coefficient estimates = - 0.21, p = 0.21). The sigmoid curve fitting the daily average number of eggs per live hen for the first 60 days in the laying barn and over the full laying barn period are represented per pen in Fig 3D and 3E, respectively. Parameters are illustrated in green in Fig 3D. We provided the parameter estimates of each curve per pen in the S2 Fig. For example, from these estimates, we can observe that the levels at which the average number of daily eggs per live hen stabilised during the first 60 days in the laying barn (as measured by the *a* estimate) for OFH pens was between 0.948 and 0.964, and slightly lower for STAN pens (0.926–0.956). The time point (i.e., number of days since transfer to the laying barn) at the inflection point of the curve (as measured by the *m* estimate)

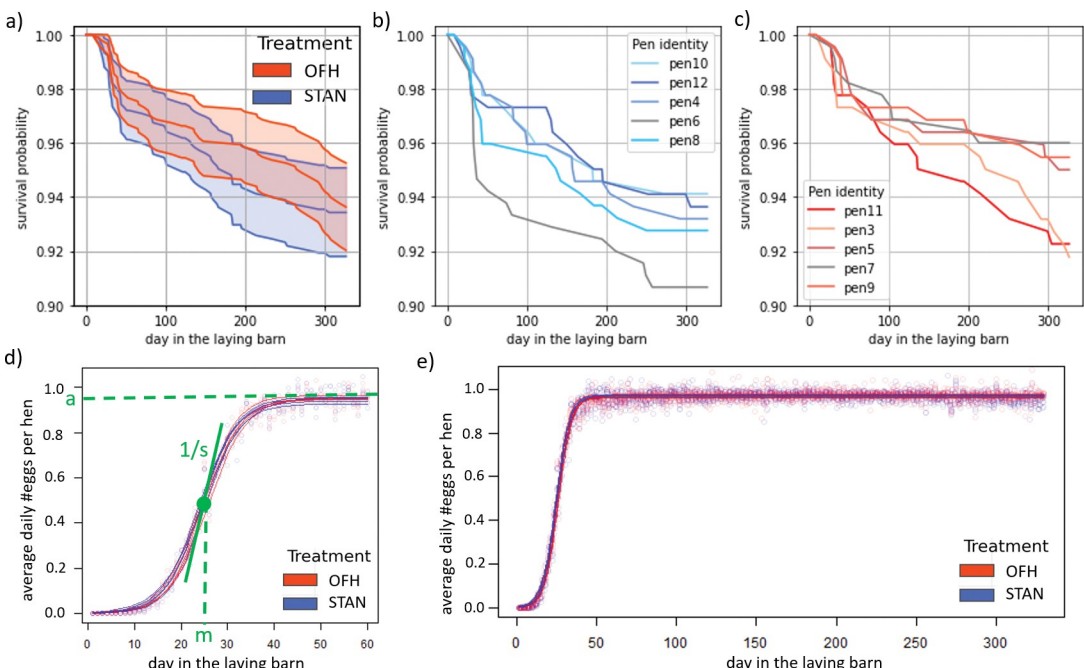

**Fig 3. Survival probability and average daily number of eggs per live hens, over time and per treatment.** Survival probability of OFH and TRAN hens over time in the laying barn (a), and per pen (STAN pens: (b) and OFH pens: (c), including the two special pens containing 205 animals of the one treatment group but also an additional 20 Lohmann LSL hens, highlighted in grey). Average daily number of eggs per hen (data points), with the fitted sigmoid curve for each pen, during the first 60 days in the laying barn (d) and the full period in the laying barn (e). Their associated parameters *a* (as an indication of the level at which egg production stabilise), *m* (as an indication of the time point at the inflection point of the curve), and *s* (steepness of the curve at a/2) are illustrated in green (d), and estimates given in the S2 Fig. Red and blue colours represent the treatment groups, OFH and STAN, respectively.

is between 24.05 and 25.79 for OFH pens and slightly earlier for STAN pens between 23.71 and 25.00. However, we found no treatment effect on any of the sigmoidal parameters (a, m, s) when considering either the first 60 days or the entire period in the laying barn.

## Discussion

In this study, we found that on average, OFH hens had more severe KBF throughout the laying period, transitioned less between the aviary tiers and spent less time on the littered floor and more on the top tier compared to STAN hens. We found no treatment effect on the daily presence in the WG, almost none on the two nestbox tier related behaviours, and none in body mass, feather damage, survival probability, and number of egg per live hens. To our knowledge, this study is the first to investigate the potential effects of the standard commercial hatchery process throughout the laying period on commercial hens' welfare and spatial behaviour, and in particular to assess the effect on KBF. These findings suggest that early-life stressors associated with the commercial hatchery environment–here, transport-related stressors and temporary post-hatch resource deprivation–may have prepared hens for their adult environment and thereby support the environmental matching hypothesis.

### Welfare indicators

According to the environmental matching hypothesis, early-life stress can prepare an individual for similar adversities encountered in adulthood [12]. The adulthood of commercial laying

hens is typically accompanied by aversive situations, including the transfer to the laying barn [34]. The potential stressors associated with this transfer, such as being transported to a new housing with new conspecifics, echo those encountered by STAN chicks. Therefore, the adult environment of commercial hens may better match with the environmental conditions experienced by day-old STAN chicks rather than those of day-old OFH chicks.

The result showing less severe KBF in STAN hens across the laying period compared to OFH hens provides support for the environmental matching hypothesis. Although KBF are considered as one of the greatest welfare issues in the egg production industry [16, 17, 35–39], effects of early-life stressors induced by commercial hatchery on KBF was to our knowledge not yet investigated. Various factors are thought to contribute to the prevalence of KBF, including genetic predisposition [40], nutrition [40], de-mineralized bone aggravated by the high egg laying rate [41–43], inactivity compromising bone health [44], and certain physical elements of the housing system that may cause pressure on the bone when hens are perching [45] or lead to trauma resulting from impact collisions [20, 44, 46, 47]. Given that the STAN and OFH hens originated from the same parent flock, it is unlikely that genetic predisposition could explain this result. Moreover, apart from the nutritional differences after hatching, where OFH chicks had immediate access to feed and water, which is unlikely to negatively impacted their bone health, both STAN and OFH hens received the same nutrition. Because we observed minor differences in egg-production during the onset of lay, which, if anything, would suggest that OFH hens had a slower onset of lay, it is also unlikely that these differences would explain the result.

Given the limited amount of genetic and environmental differences between treatment groups outside of the treatment specific differences, we believe behavioural differences in activity and/or different usage of the housing system could explain why OFH hens had more severe KBF compared to STAN hens. For instance, our finding indicating a 31% reduction in vertical travelled distance during the first month for hens hatched on farms, may reflect a more general pattern of inactivity among OFH hens which could lead to poorer bone health and in turn higher KBF [44]. Additionally, OFH hens spent more time on the top tier, where hard metal perches are more abundant. High perch use would result in overall greater pressure on the keel bone leading to deviated keels, which in turn can weaken the keel's overall structural integrity [46]. Overall, we cannot exclude the possibility that the treatment difference we observed in the severity of KBF may be an indirect effect of the differences we observed in spatial behaviours between the treatment groups. Further research is required to replicate this finding and to understand the underlying mechanisms involved.

More recent evidence suggested that factors related to egg production, including internal pressure during the egg-laying process [48, 49] and an early onset of lay [50], could favour KBF. We found no evidence of treatment effect in the average number of egg per live hens, nor did we observe treatment effect on the timing of the onset of lay. In fact, a previous study showed that on-farm hatched hens had a slower onset of egg-laying than hens hatched in commercial hatchery at 15–20 weeks of age [2]. Hence, while an early onset of laying may not explain the treatment effect on KBF observed in our study, it is plausible that other factors related to the internal pressure during the egg-laying process could be operating in the current effort.

We found that OFH hens had on average a similar body mass to STAN hens beginning with our earliest measurement at 7 DOA. We expected OFH hens to weigh more because of direct access to feed after hatching, which previous studies on day-old chicks demonstrated [3, 8, 9], but also because it is possible that animals may gain less weight due to early life stress [51, 52]. However, beyond one day of age, the effect in previous literature is ambiguous. Studies contrasting on-farm hatching to standard commercial practices found no treatment

differences at 4 or 7 DOA [3, 8], a tendency for on-farm hatched chicks to weigh more up to 11 weeks of age [9], and that on-farm hatched chicks weighed less at 8, 15, 22 and 29 DOA [2]. Our study provides more evidence that treatment effects of body mass are not present during lay.

Furthermore, OFH hens had similar feather damage to STAN hens, which supports neither the silver-spoon nor the environmental matching hypothesis, but is in line with the inconclusive findings in earlier literature that reported both positive [2] and negative [8] effects. The ambiguity surrounding these results may stem from the multifactorial nature of feather damage, which includes factors such as feather pecking and abrasion resulting from different parts of the structures [53]. Therefore, the influences of the physical and social environments on feather damage could outweigh or interact with effects from early-life stressors.

## Spatial behaviours

The literature on behavioural differences between hens that hatched on-farm versus in a commercial hatchery is sparse and mainly conducted in test arenas [2, 3, 54–56], hindering the extrapolation of the results to commercial settings, typically characterized by more complex housing systems and larger groups. In this study, we used tracking technology to monitor movements of hens within a quasi-commercial aviary system throughout the laying period.

We found that for the first three months post-transfer to the laying barn (up to 7-month-old) STAN hens spent more time on the littered floor and less on the top tier. The top tier has been shown to be used more extensively by hens with more severe keel bone fractures [57] and throughout full days following their transfer to the laying barn [21] (in a previous study on the same hens). Thus, it is possible that hens use this area over the day to offset stress or pain. Unlike the top tier, the littered floor promotes a diverse range of natural behaviours, including locomotion, dust bathing, exploring, foraging, and scratching. Therefore, these findings could indicate that OFH hens exhibited fewer natural behaviours compared to STAN hens in this early laying barn period. Furthermore, as OFH hens became behaviourally more similar to STAN hens over time (Table 2), these results further suggest that OFH hens required a prolonged period to establish their daily behavioural patterns. Overall, STAN hens may have exhibited better abilities in coping with the transfer to the laying barn, which would provide further support for the environmental matching hypothesis. Early-life stress can enhance behavioural flexibility [58], improve stress coping later in life [59], and facilitate spatial learning and memory [58, 60], especially when experienced close in time and within the same context that is encountered later [61]. Therefore, increased stress induced by the commercial hatchery process near the time of transfer to the rearing barn could have induced focused attention and improved the memory of relevant information allowing STAN hens to better cope to the laying barn.

In addition to the potential effect of early-life stressors, it is possible that the STAN chicks may have exhibited faster acclimatization to their new environment due to the inherent effects of their transition from the hatchery to the farm. As suggested by Skånberg et al. [62], it is possible that environmental change during rearing plays a role in enhancing layer chicks' adaptability later in life. Specifically, STAN hens may have learned appropriate cognitive and locomotive skills during the rearing period, facilitating their adaptation to the aviary system. However, a previous study comparing the cognitive ability of layer chicks that had temporary post-hatch resource deprivation and eight-hour transport, to those that had ad libitum access to feed and water and were not transported, found no treatment effect in a cognitive test [9]. Further research is needed to investigate whether STAN hens possess better spatial-cognitive abilities in the laying period.

It is also possible that the additional early-life stressors experienced by STAN hens could have altered the functional and structural development of the Hypothalamic-Pituitary-Adrenal (HPA)-axis [63, 64]. Typically, stress induced in the early postnatal period results in HPA-axis hyper-reactivity during adulthood [63–66], with enhanced depression-like behaviours and anxiety [67]. Previous studies on laying hens generally supports such alterations of the HPA-axis. It was shown that hens from a commercial hatchery had a more sensitive HPA-axis and a stronger reaction to stressors during the first weeks of life compared to on-farm hatched hens [2, 55]. Here, we showed that STAN hens spent more time on the littered floor, less time on the top tier, and exhibited their typical behavioural patterns earlier after the transfer to the laying barn, in comparison to OFH hens. We believe that these behavioural pattern are not indicative of a stronger reaction to the transfer, and therefore believe that the additional early-life stressors experienced by STAN hens may not have significantly impacted their HPA-axis. Alternatively, it is also possible that the subsequent adversities encountered by all chicks in the rearing barn could have triggered the HPA axis of OFH hens to reach conditions resembling that of the STAN hens [63]. However, further research is necessary to understand how transportation and early life conditions influence the HPA-axis and its relationship to movement in laying hens.

We found almost no treatment effect on the two nestbox tier behaviours, which supports the idea of limited behavioural plasticity related to the use of the nestbox tier. Given the strong human selection for high productivity and the high motivation to use nestboxes [68], it is possible that those behaviours are more tightly correlated with physiology or with strong animal needs and thus are less plastic than others. That is, these behaviours may be less influenced by external or internal factors and more repeatable across different contexts, as suggested by previous findings [57, 69].

### Production traits

Until *in ovo* sexing becomes practical at commercial scale, differences between hatchery and on-farm treatments may not have practical effects relevant to animal welfare. However, with the advancement of *in ovo* sexing techniques [70], it is possible that on-farm hatching practices will become a standard method in the future. Therefore, it is important to compare both animal welfare and productivity between commercial hatchery and on-farm treatments within commercially relevant settings. We found no treatment effect on mortality, despite on-farm practices have already been shown to reduce total mortality in broilers [5]. We also found no treatment effect on the average number of egg per live hens in terms of both the level at which it stabilized (i.e., parameter *a*) and the time point at which the curve reached its inflection point (i.e., parameter *m*). To our knowledge, only two studies [2, 8] have compared egg-production between such treatments, by collecting daily production data at the pen-level. The authors found that OFH hens had a slower onset of egg laying than STAN hens at 15–20 weeks of age [2] in one study and laid more and bigger eggs at 19–25 weeks of age in the second [8]. The absence of a statistical difference in our study may be attributed to the limited sample size and further research with substantially more pens or at the individual-level is needed to determine whether there are differences in egg-production between on-farm hatched hens and those hatched in commercial hatcheries.

### Limitations

This study aimed to assess the potential impact of commercial hatchery practices and potential benefits of on-farm hatching for animal welfare within commercially relevant settings throughout most of the production period. Some potential early life stressors were in this

study uniquely experienced by STAN animals, including food and water deprivation [71] and the transportation at one day of age to a new environment [34]. Yet, the hatchery-related procedures applied to OFH chicks might also have been aversive, including the transportation at 18 days of incubation [72] and the vaccination and sexing at one day of age. We designed the experiment so that all hatchery-related procedures should be less, or when not possible, equally, aversive for the OFH treatment compared to the TRAN treatment. However, our methodology did not allow to determine if the increased cumulative adversities encountered by STAN chicks during their development led to higher stress responses than those seen in OFH chicks. Hence, to assess the relative benefit of on-farm hatching practices for practical applications, future research should compare physiological stress and cognitive responses between treatment in the rearing and laying phase and strive to replicate on-farm hatching procedures more closely to the envisioned future practices (e.g., integrating *in ovo* sexing for the OFH treatment).

## Conclusion

Hens from both treatments originated from the same parent flock yet produced different phenotypes depending on their early-life environments, suggesting the presence of developmental plasticity in our commercial hens. By providing nuance to the relative benefit of on-farm hatching for animal welfare described by previous literature, this study highlighted the critical role in designing early-life environments in accordance with the adult environment. For laying hens housed in aviary systems, our results could suggest that hatching on-farm rather than in commercial hatchery may lead to an increased mismatch between the hatching and laying environments that can be detrimental to animal welfare. We conclude that future research is needed to determine whether conventional husbandry practices, originally designed for hens hatched in commercial hatcheries, need to be adapted for hens hatched on-farm, such as implementing smoother transfers between rearing to laying barns.

## Supporting information

**S1 Text. More/less explorer class.**
(PDF)

**S1 Fig. Eggshell temperature over time.** We monitored 30 OFH eggs every six hours, for a total of 7 timestamps, until a significant proportion of the chicks hatched.
(PNG)

**S2 Fig. Parameter estimates of the sigmoid curve per pen.** Parameter *a* may be viewed as an indication of the level at which egg production stabilise. Parameter *m* as an indication of the time point at the inflection point of the curve. These estimates of the curves fitting the first 60 days and the full period in the laying barn are given in (b) and (d), respectively.
(PNG)

**S1 Table. Hatching rate over time for the OFH chicks.**
(PDF)

**S2 Table. Estimated marginal means and 95% confidence intervals for welfare indicators.**
(PDF)

**S3 Table. Model outputs for welfare indicators.**
(PDF)

**S4 Table. Bootstrapped estimates and p-values for the model fitting spatial behaviours.** (PDF)

## Acknowledgments

We are thankful to the Aviforum staff for animal care and for collecting the daily mortality as well as the egg-production data. We would also like to thank two anonymous reviewers and Tom Smulders for valuable feedback on a previous version of this manuscript.

## Author Contributions

**Conceptualization:** Camille M. Montalcini, Matthew B. Petelle, Michael J. Toscano.

**Data curation:** Camille M. Montalcini.

**Formal analysis:** Camille M. Montalcini.

**Funding acquisition:** Michael J. Toscano.

**Investigation:** Camille M. Montalcini, Matthew B. Petelle.

**Methodology:** Camille M. Montalcini, Matthew B. Petelle.

**Project administration:** Michael J. Toscano.

**Resources:** Michael J. Toscano.

**Supervision:** Matthew B. Petelle, Michael J. Toscano.

**Visualization:** Camille M. Montalcini.

**Writing – original draft:** Camille M. Montalcini.

**Writing – review & editing:** Camille M. Montalcini, Matthew B. Petelle, Michael J. Toscano.

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
