## [Decision Letter · Decision Letter 0]

6 Oct 2023

PONE-D-23-28828Commercial hatchery practices have long-lasting effects on laying hens’ spatial behaviour and healthPLOS ONE

Dear Dr. Montalcini,

Thank you for submitting your manuscript to PLOS ONE. After careful consideration, we feel that it has merit but does not fully meet PLOS ONE’s publication criteria as it currently stands. Therefore, we invite you to submit a revised version of the manuscript that addresses the points raised during the review process.

Dear Dr., Camille M. Montalcini

Thank you for submitting your manuscript to PLOS ONE. After careful consideration, we have decided that your manuscript needs Major Revision.

Kind regards,

Prof. Lamiaa Mostafa Radwan, Ph.D.

Academic Editor

PLOS ONE

Editor Comments:

1- Material and methods need more details, especially the method used to determine the sex of the newly hatched chicks.

2- The statistical model used needs to be reviewed and written with more clarification and detail, and all data in the manuscript must be subjected to statistical analysis.

**Reviewer1**

A very interesting paper on an important topic. I would like the authors to attend to the following comments and suggestions:

- The eggs were transported on day 18 of incubation to the on-farm hatch system. Please consider the possible stress this may have induced on the embryos. Recent research has showed that the embryos may be susceptible to stress at this point and this may have affected your results (Nordquist, R.E., Vernooij, J.C.M., Dull, C.L., Pascual, A., van der Linde, G., Goerlich, V.C., 2022. The effects of transport of 18-day old hatching eggs on physiology and behaviour of slow growing broiler chicken. Applied Animal Behaviour Science 257.)

- The newly hatched chicks were manually sexed - please provide more information on how this was done and how this may possibly have affected the chicks that remained in the experiment.

- I think the egg production data should in fact be possible to test in a more rigorous statistical manner. Although I appreciate that there were only five pens per treatment, a statistical evaluation would be helpful in order to validate the findings.

- The differences between treatment groups in KBF are surprising (to me at least) and I think the authors should consider the possibility that the early treatment in the OFH may have contributed (egg transport, sexing, vaccination).

- In general, this is a commendable paper that should be published after the authors have considered my comments. However, it should also be noted that until in-ovo sexing is a feasible standard method, the differences between the hatchery treatments and the OFH may not be very large from the perspective of animal welfare

.

**Reviewer2**

Respected Authors,

The idea is very well conceptualized, and manuscript is written well, however, at this position I suggest some corrections.

I have mentioned all my suggestions/corrections in attached file, kindly follow the instructions.

**Reviewer3**

This is an interesting study that shows that on-farm hatching does not necessarily always result in improved welfare for the laying hens for the rest of their lives. In fact, it seems like traditionally hatched birds that have experienced transport in early life cope better with the move from rearing barn to laying barn. It is nice to see results honestly presented, even though I assume the authors started from the hypothesis that on-farm hatching would be beneficial (given the rest of the literature).

I don't have any major issues with this paper. The results seem clear, as do the methods. I do have a few requests for clarification, though, and a (very) few editing suggestions.

1) Could the authors please explain a bit better the implications of the fact that some birds were killed and then replaced by other animals? Does this mean that the sample of OFH and STAN birds at different time points were partially overlapping, but not identical groups of birds? Does that have any implications for the stats?

2) In the statistics, often different data structures were chosen for different variables. Please explain how you decided on each of these structures and why.

3) Given the recently presented results on the difference beteen KBF high on the keel (typically on the outside, and from collisions) vs. low on the keel (typically on the inside and due to egg laying), do the authors have any information on whether the difference between OFH birds and STAN birds was mostly due to "collision breaks" or to "egg laying breaks"? The discussion suggests they believe collisions are more to blame due to inexperience with moving around the aviary, but it would be good to see the other discussed.

4) I don't quite follow the reasoning on line 395 about counterbalancing. The study did not really assess HPA axis activity, so what are the authors trying to explain away here, then?

5) The authors don't report this, but I wonder if there was any effect of OFH on mortality in the laying barn (and therefore on total egg production, rather than per-bird production).

a. line 140 "between 2 and 8 o'clock" instead of "2 and 8 hour"? Or do you mean between 2 and 8 hours after lights came on?

b. line 153: "KBF severity and feather damage" instead of "KBF severity the feather damage"?

c. line 234: although what you say is not wrong, the fact that red is left of blue in the figure makes it more logical to write "Red and blue colours represent the treatment groups OFH and STAN respectively", as they are then in the same order as in the figure.

d. line 238: "Model coefficients" instead of "Models' coefficients"

e. line 239: "To help further interpret" instead of "To help further interpreting"

f. line 342-343: please clarify the phrase "potential decreased weight gained" - I get confused about what goes up and what goes down.

We look forward to receiving your revised manuscript.

Kind regards,

Lamiaa Mostafa Radwan, Ph.D.

Academic Editor

PLOS ONE

Journal Requirements:

Additional Editor Comments:

Dear Dr., Camille M. Montalcini

Thank you for submitting your manuscript to PLOS ONE. After careful consideration, we have decided that your manuscript needs Major Revision.

Kind regards,

Prof. Lamiaa Mostafa Radwan, Ph.D.

Academic Editor

PLOS ONE

Editor Comments:

1- Material and methods need more details, especially the method used to determine the sex of the newly hatched chicks.

2- The statistical model used needs to be reviewed and written with more clarification and detail, and all data in the manuscript must be subjected to statistical analysis.

Reviewer1

A very interesting paper on an important topic. I would like the authors to attend to the following comments and suggestions:

- The eggs were transported on day 18 of incubation to the on-farm hatch system. Please consider the possible stress this may have induced on the embryos. Recent research has showed that the embryos may be susceptible to stress at this point and this may have affected your results (Nordquist, R.E., Vernooij, J.C.M., Dull, C.L., Pascual, A., van der Linde, G., Goerlich, V.C., 2022. The effects of transport of 18-day old hatching eggs on physiology and behaviour of slow growing broiler chicken. Applied Animal Behaviour Science 257.)

- The newly hatched chicks were manually sexed - please provide more information on how this was done and how this may possibly have affected the chicks that remained in the experiment.

- I think the egg production data should in fact be possible to test in a more rigorous statistical manner. Although I appreciate that there were only five pens per treatment, a statistical evaluation would be helpful in order to validate the findings.

- The differences between treatment groups in KBF are surprising (to me at least) and I think the authors should consider the possibility that the early treatment in the OFH may have contributed (egg transport, sexing, vaccination).

- In general, this is a commendable paper that should be published after the authors have considered my comments. However, it should also be noted that until in-ovo sexing is a feasible standard method, the differences between the hatchery treatments and the OFH may not be very large from the perspective of animal welfare

.

Reviewer2

Respected Authors,

The idea is very well conceptualized, and manuscript is written well, however, at this position I suggest some corrections.

I have mentioned all my suggestions/corrections in attached file, kindly follow the instructions.

Reviewer3

This is an interesting study that shows that on-farm hatching does not necessarily always result in improved welfare for the laying hens for the rest of their lives. In fact, it seems like traditionally hatched birds that have experienced transport in early life cope better with the move from rearing barn to laying barn. It is nice to see results honestly presented, even though I assume the authors started from the hypothesis that on-farm hatching would be beneficial (given the rest of the literature).

I don't have any major issues with this paper. The results seem clear, as do the methods. I do have a few requests for clarification, though, and a (very) few editing suggestions.

1) Could the authors please explain a bit better the implications of the fact that some birds were killed and then replaced by other animals? Does this mean that the sample of OFH and STAN birds at different time points were partially overlapping, but not identical groups of birds? Does that have any implications for the stats?

2) In the statistics, often different data structures were chosen for different variables. Please explain how you decided on each of these structures and why.

3) Given the recently presented results on the difference beteen KBF high on the keel (typically on the outside, and from collisions) vs. low on the keel (typically on the inside and due to egg laying), do the authors have any information on whether the difference between OFH birds and STAN birds was mostly due to "collision breaks" or to "egg laying breaks"? The discussion suggests they believe collisions are more to blame due to inexperience with moving around the aviary, but it would be good to see the other discussed.

4) I don't quite follow the reasoning on line 395 about counterbalancing. The study did not really assess HPA axis activity, so what are the authors trying to explain away here, then?

5) The authors don't report this, but I wonder if there was any effect of OFH on mortality in the laying barn (and therefore on total egg production, rather than per-bird production).

a. line 140 "between 2 and 8 o'clock" instead of "2 and 8 hour"? Or do you mean between 2 and 8 hours after lights came on?

b. line 153: "KBF severity and feather damage" instead of "KBF severity the feather damage"?

c. line 234: although what you say is not wrong, the fact that red is left of blue in the figure makes it more logical to write "Red and blue colours represent the treatment groups OFH and STAN respectively", as they are then in the same order as in the figure.

d. line 238: "Model coefficients" instead of "Models' coefficients"

e. line 239: "To help further interpret" instead of "To help further interpreting"

f. line 342-343: please clarify the phrase "potential decreased weight gained" - I get confused about what goes up and what goes down.

Reviewers' comments:

Reviewer's Responses to Questions

**Comments to the Author**

1. Is the manuscript technically sound, and do the data support the conclusions?

Reviewer #1: Yes

Reviewer #2: Partly

Reviewer #3: Yes

2. Has the statistical analysis been performed appropriately and rigorously? 

Reviewer #1: Yes

Reviewer #2: Yes

Reviewer #3: Yes

3. Have the authors made all data underlying the findings in their manuscript fully available?

Reviewer #1: Yes

Reviewer #2: Yes

Reviewer #3: Yes

4. Is the manuscript presented in an intelligible fashion and written in standard English?

Reviewer #1: Yes

Reviewer #2: Yes

Reviewer #3: Yes

5. Review Comments to the Author

Reviewer #1: A very interesting paper on an important topic. I would like the authors to attend to the following comments and suggestions:

- The eggs were transported on day 18 of incubation to the on-farm hatch system. Please consider the possible stress this may have induced on the embryos. Recent research has showed that the embryos may be susceptible to stress at this point and this may have affected your results (Nordquist, R.E., Vernooij, J.C.M., Dull, C.L., Pascual, A., van der Linde, G., Goerlich, V.C., 2022. The effects of transport of 18-day old hatching eggs on physiology and behaviour of slow growing broiler chicken. Applied Animal Behaviour Science 257.)

- The newly hatched chicks were manually sexed - please provide more information on how this was done and how this may possibly have affected the chicks that remained in the experiment.

- I think the egg production data should in fact be possible to test in a more rigorous statistical manner. Although I appreciate that there were only five pens per treatment, a statistical evaluation would be helpful in order to validate the findings.

- The differences between treatment groups in KBF are surprising (to me at least) and I think the authors should consider the possibility that the early treatment in the OFH may have contributed (egg transport, sexing, vaccination).

- In general, this is a commendable paper that should be published after the authors have considered my comments. However, it should also be noted that until in-ovo sexing is a feasible standard method, the differences between the hatchery treatments and the OFH may not be very large from the perspective of animal welfare.

Reviewer #2: Respected Authors,

The idea is very well conceptualized, and manuscript is written well, however, at this position I suggest some corrections.

I have mentioned all my suggestions/corrections in attached file, kindly follow the instructions.

Reviewer #3: This is an interesting study that shows that on-farm hatching does not necessarily always result in improved welfare for the laying hens for the rest of their lives. In fact, it seems like traditionally hatched birds that have experienced transport in early life cope better with the move from rearing barn to laying barn. It is nice to see results honestly presented, even though I assume the authors started from the hypothesis that on-farm hatching would be beneficial (given the rest of the literature).

I don't have any major issues with this paper. The results seem clear, as do the methods. I do have a few requests for clarification, though, and a (very) few editing suggestions.

1) Could the authors please explain a bit better the implications of the fact that some birds were killed and then replaced by other animals? Does this mean that the sample of OFH and STAN birds at different time points were partially overlapping, but not identical groups of birds? Does that have any implications for the stats?

2) In the statistics, often different data structures were chosen for different variables. Please explain how you decided on each of these structures and why.

3) Given the recently presented results on the difference beteen KBF high on the keel (typically on the outside, and from collisions) vs. low on the keel (typically on the inside and due to egg laying), do the authors have any information on whether the difference between OFH birds and STAN birds was mostly due to "collision breaks" or to "egg laying breaks"? The discussion suggests they believe collisions are more to blame due to inexperience with moving around the aviary, but it would be good to see the other discussed.

4) I don't quite follow the reasoning on line 395 about counterbalancing. The study did not really assess HPA axis activity, so what are the authors trying to explain away here, then?

5) The authors don't report this, but I wonder if there was any effect of OFH on mortality in the laying barn (and therefore on total egg production, rather than per-bird production).

a. line 140 "between 2 and 8 o'clock" instead of "2 and 8 hour"? Or do you mean between 2 and 8 hours after lights came on?

b. line 153: "KBF severity and feather damage" instead of "KBF severity the feather damage"?

c. line 234: although what you say is not wrong, the fact that red is left of blue in the figure makes it more logical to write "Red and blue colours represent the treatment groups OFH and STAN respectively", as they are then in the same order as in the figure.

d. line 238: "Model coefficients" instead of "Models' coefficients"

e. line 239: "To help further interpret" instead of "To help further interpreting"

f. line 342-343: please clarify the phrase "potential decreased weight gained" - I get confused about what goes up and what goes down.

6. PLOS authors have the option to publish the peer review history of their article (what does this mean?). If published, this will include your full peer review and any attached files.

Reviewer #1: No

Reviewer #2: No

Reviewer #3: **Yes: **Tom V. Smulders

---

## [Author Response · Author response to Decision Letter 0]

9 Nov 2023

Dear Prof. Lamiaa Mostafa Radwan and Reviewers,

Thank you very much for giving us the opportunity to revise our manuscript. We also thank the three reviewers for their kind comments and constructive suggestions. We believe their recommendations have significantly enhanced the statistical analysis, particularly in the production sub-section. It now includes a survival analysis based on the daily mortality and a statistical evaluation of the treatment's effect on egg production data. Thanks to reviewers’ suggestions, we also clarified different aspects of the Material and Methods, including the study design and the statistical analysis. The discussion now contains more detailed interpretations and further alternatives explanations (including how factors during egg-laying may relate to our results on keel bone fractures). We provide point-by-point responses in an attached word document file, and hope the reviewers’ concerns have been addressed.

With best regards,

Camille Montalcini

---

## [Decision Letter · Decision Letter 1]

27 Nov 2023

Commercial hatchery practices have long-lasting effects on laying hens’ spatial behaviour and health

PONE-D-23-28828R1

Dear Dr.Camille M. Montalcini

We’re pleased to inform you that your manuscript has been judged scientifically suitable for publication and will be formally accepted for publication once it meets all outstanding technical requirements.

Kind regards,

Lamiaa Mostafa Radwan, Ph.D.

Academic Editor

PLOS ONE

Additional Editor Comments (optional):

Dear Dr., Camille M. Montalcini

Thank you for submitting your manuscript to PLOS ONE. After careful consideration, we have decided that your manuscript needs to be Accepted.

Kind regards,

Prof. Lamiaa Mostafa Radwan, Ph.D.

Academic Editor

PLOS ONE

Kind regards,

Reviewer

Thank you for the revisions, which I think have addressed all my previous concerns. Congratulations to a very interesting piece of research.

.

.

Reviewer2

Manuscript is now acceptable from myside.

Reviewer3

Accepted

Reviewers' comments:

Reviewer's Responses to Questions

**Comments to the Author**

1. If the authors have adequately addressed your comments raised in a previous round of review and you feel that this manuscript is now acceptable for publication, you may indicate that here to bypass the “Comments to the Author” section, enter your conflict of interest statement in the “Confidential to Editor” section, and submit your "Accept" recommendation.

Reviewer #1: All comments have been addressed

Reviewer #2: All comments have been addressed

Reviewer #3: All comments have been addressed

2. Is the manuscript technically sound, and do the data support the conclusions?

Reviewer #1: Yes

Reviewer #2: Yes

Reviewer #3: Yes

3. Has the statistical analysis been performed appropriately and rigorously? 

Reviewer #1: Yes

Reviewer #2: Yes

Reviewer #3: Yes

4. Have the authors made all data underlying the findings in their manuscript fully available?

Reviewer #1: Yes

Reviewer #2: (No Response)

Reviewer #3: Yes

5. Is the manuscript presented in an intelligible fashion and written in standard English?

Reviewer #1: Yes

Reviewer #2: Yes

Reviewer #3: Yes

6. Review Comments to the Author

Reviewer #1: Thank you for the revisions, which I think have addressed all my previous concerns. Congratulations to a very interesting piece of research.

Reviewer #2: (No Response)

Reviewer #3: (No Response)

7. PLOS authors have the option to publish the peer review history of their article (what does this mean?). If published, this will include your full peer review and any attached files.

Reviewer #1: No

Reviewer #2: No

Reviewer #3: **Yes: **Tom V. Smulders

---

## [Editor Report · Acceptance letter]

11 Dec 2023

PONE-D-23-28828R1 

Commercial hatchery practices have long-lasting effects on laying hens’ spatial behaviour and health 

Dear Dr. Montalcini:

I'm pleased to inform you that your manuscript has been deemed suitable for publication in PLOS ONE. Congratulations! Your manuscript is now with our production department. 

Kind regards, 

on behalf of

Prof. Dr. Lamiaa Mostafa Radwan 

Academic Editor

PLOS ONE